# A comprehensive protocol for PDMS fabrication for use in cell culture

**Aisling J. Greaney**[1,2], **Clíona M. McCarthy**[1,2], **Jishnu Padacherri Vethil**[1,2],
**Mannthalah Abubaker**[1,2], **Erin C. Reardon**[1,2], **Frederick D. Crowley**[1,2],
**Eoghan M. Cunnane**[1,2], **John J. E. Mulvihill**[1,2]*

**1** School of Engineering, University of Limerick, Limerick, Ireland, **2** Bernal Institute, University of Limerick, Limerick, Ireland

* John.mulvihill@ul.ie

## Abstract

Cells exhibit remarkable sensitivity to the mechanical properties of their surrounding matrix, particularly stiffness changes, a phenomenon known as cellular mechano-transduction. In vivo, tissues exhibit a wide range of stiffness, from kilopascals (kPa) to megapascals (MPa), which can alter with aging and disease. Traditional cell culture methods employ plastic substrates with stiffness in the gigapascal range, which does not accurately mimic the physiological conditions of most biological tissues. Therefore, employing substrates that can be engineered to span a wide range of stiffnesses, closely resembling the native tissue environment, is crucial for obtaining results that more accurately reflect cellular responses in vivo. Polydimethylsiloxane (PDMS) substrates are widely used in cell culture for their ability to simulate tissue stiffness, but their optimization presents several challenges. Fabrication requires precise control over mixing, weighing, and curing to ensure reproducible mechanical properties. Inconsistent preparation can lead to improperly cured PDMS substrates, compromising experimental outcomes. Additionally, PDMS's inherent hydrophobicity poses challenges for cell attachment, necessitating surface modifications to enhance adhesion. Moreover, the risk of contamination during the sterilization process necessitates stringent protocols to maintain cell culture integrity. These challenges are further compounded by substrate autofluorescence which can cause difficulties when imaging cells. The aim of this study is to develop a standardized method for fabricating PDMS substrates with tuneable stiffness, ranging from kPa to MPa, suitable for diverse cell types using standard laboratory equipment. This method aims to minimize the complexity and equipment required for PDMS fabrication, ensuring reproducibility and ease of use. Achieving consistent and contaminant-free PDMS substrates will facilitate a broader adoption of these substrates in mechanobiology research and improve the relevance of in vitro models to in vivo conditions. Ultimately, contributing to a more comprehensive understanding of cellular responses to mechanical cues in health and disease.

**Data availability statement:** All relevant data are within the manuscript and its Supporting Information files.

**Funding:** The research conducted in this publication was funded by the Irish Research Council under grant number GOIPG/2021/1433 The funders had no role in study design, data collection and analysis, decision to publish, or preparation of the manuscript

**Competing interests:** The authors have declared that no competing interests exist.

## Introduction

The structure and stiffness of a tissue's extracellular matrix regulate cellular function; therefore, changes in its stiffness can alter the mechanosensitivity of these resident cells [1–6]. Understanding how cells respond to mechanical cues, particularly stiffness alterations, is crucial for advancing knowledge in cell biology and disease mechanisms. Traditional cell culture practices typically use plastic culture ware with stiffness values in the gigapascal range [7]. However, this does not accurately represent the mechanical properties of most biological tissues, which range from kPa to MPa [8,9]. This discrepancy can lead to misleading conclusions about cellular behaviour [3,10–12]. R.G. Wells [11] emphasized this concern, noting that "in vitro model systems that involve culturing cells on tissue culture plastic will need to be reexamined and may be found inappropriate for studying some cell behaviors". Hence, there is a pressing need to transition from plastic to substrates that can be engineered to more accurately mimic the physiological stiffness of tissues.

Numerous studies have shown that cells are highly mechanosensitive, responding distinctly to changes in substrate stiffness [4,9,11–19]. Specifically, substrate stiffness significantly influences both stem cell differentiation and cancer cell behaviour. For instance, human mesenchymal stem cells differentiate into neuronal lineages on softer substrates and into osteogenic lineages on stiffer substrates [15]. Moreover, cancer cells exhibit altered behaviours in response to substrate stiffness, such as changes in proliferation, migration, and drug resistance [16,17]. Therefore, it is crucial to understand the impact of substrate stiffness on the cells under investigation. Future studies should focus on developing substrates that facilitate the transition from conventional plastic culture ware to those with tuneable stiffness, ranging from kPa to MPa.

In cell culture, a variety of substrates are commonly employed to investigate cellular responses to mechanical properties, including polyacrylamide hydrogels, gelatin methacryloyl, collagen gels, Matrigel, polyethylene glycol hydrogels, alginate hydrogels, and PDMS [20–32]. Among these, PDMS is particularly favoured due to its tuneable stiffness, biocompatibility, optical transparency, ease of fabrication, surface modifiability, and cost-effectiveness [5,20–22,33,34]. Notably, PDMS-based elastomers, such as Sylgard 184 and Sylgard 527, are widely used in biological and engineering research due to their tuneable stiffness, ranging from the kilopascal (kPa) to the megapascal (MPa) scale, which allow them to be used as a mimetic of various biological tissue types. These differences arise from variations in polymer chain lengths and crosslinking densities, allowing researchers to modulate substrate mechanics while maintaining consistent surface chemistry [22]. This flexibility supports diverse applications; Sylgard 184, with a modulus in the MPa range, is suited for culturing cells that reside in rigid environments, such as fibroblasts, endothelial cells, and osteoblasts [35–38]. It is also widely used in microfluidics, soft lithography, and electronic encapsulation due to its durability and chemical resistance [39–41]. Sylgard 527, with a kPa-range modulus, is ideal for softer substrates used in stem cell and myoblast cultures [42–44] and is suitable for sealing and protecting various electronic devices, especially those with delicate components [45].

Studies utilizing these elastomers have demonstrated significant findings, such as T lymphocytes exhibiting biphasic spreading responses to stiffness increases [5], muscle and nerve cells showing morphological and functional changes in response to varying substrate stiffness [22], and corneal epithelial cells adhering and proliferating better on substrates mimicking natural corneal stiffness [21]. These insights highlight the critical role of substrate stiffness in influencing cell behaviour and underscore the utility of PDMS-based elastomers in mechanobiology research. By using these PDMS-based elastomers, researchers could mimic different tissue environments and study how various cell types respond to mechanical cues. However, despite extensive research, no study has standardized a method for fabricating substrates with tuneable stiffness, from kPa to MPa, suitable for various cell types using standard laboratory equipment.

Fabricating PDMS substrates for cell culture presents significant challenges related to mechanical stability, biocompatibility, and usability, which are often inadequately addressed in the existing literature. Achieving consistent mechanical properties requires precise control over mixing, weighing, and curing processes, as minor deviations can alter PDMS stiffness and impact cellular behaviour. Sylgard 184 exhibits an increase in stiffness with higher curing temperatures due to enhanced crosslinking, with its elastic modulus ranging from ~1 to ~4 MPa between 25°C and 200°C [46,47]. In contrast, Sylgard 527 remains highly compliant, with negligible changes in elastic modulus, maintaining values within the low kPa range regardless of curing temperature [21,48]. Biocompatibility necessitates rigorous sterilization methods such as ethanol immersion, UV irradiation, or oxygen plasma treatment to prevent contamination and ensure suitability for cell growth [49–52]. Addressing the inherent hydrophobicity of PDMS surfaces is crucial for promoting cell adhesion and spreading, often requiring surface modifications like oxygen plasma or UV/ozone treatment, chemical functionalization, or coating with hydrophilic polymers or biomolecules [33,34,53–63]. Usability concerns include overcoming substrate autofluorescence during immunofluorescent analysis and accommodating diverse cell types, from primary cells to established lines, which is essential for broad applicability in research. Access to specialized equipment for PDMS fabrication further complicates these challenges. Many studies rely on specialized mixers for combining the silicone base (Part A) and the curing agent (Part B) of PDMS-based elastomers, oxygen plasma treaters for making the PDMS surface hydrophilic, and spin coaters for preparing substrates suitable for fluorescent imaging [20–22,34]. Despite extensive literature discussing different fabrication techniques, a unified protocol encompassing all challenges associated with PDMS substrate preparation for cell culture remains absent. Therefore, the development of a simple, reproducible protocol that can be easily implemented in any laboratory setting is necessary.

The aim of this study is to develop a standardized method for fabricating PDMS substrates with tuneable stiffness ranging from kPa to MPa and to systematically address the associated challenges. Analysing cells on substrates with varying stiffness is crucial because it provides insights into how mechanical properties influence cellular behaviour. This knowledge is essential for the development of biomaterials with tailored mechanical properties for specific biomedical applications and for understanding the mechanistic underpinnings of diseases, potentially unveiling novel therapeutic targets.

## Materials and methods

The protocol in this peer-reviewed article is published on protocols.io https://doi.org/10.17504/protocols.io.36wgqnn7xgk5/v1 and is included for printing as a S1 File with this article.

A printable version of this protocol is available as S2 File with this article and a process flow on this protocol is available to download as S1 Fig for more information.

## Expected results

The development of an effective protocol for preparing PDMS substrates for cell culture applications is anticipated to meet several critical criteria. Each criterion is essential to ensure the mechanical stability, biocompatibility, and usability of the PDMS substrates in various experimental contexts.

## Mechanical stability

The mechanical stability criterion ensures that the PDMS substrates maintain their mechanical properties and geometrical consistency over time, which is crucial for reproducibility and reliability in cell culture experiments. Achieving highly consistent substrate stiffness and thickness across all prepared concentrations is expected. Consistency of the substrate is crucial as variations in stiffness and thickness can lead to significant differences in cell behaviour, such as adhesion, proliferation, and differentiation, making it difficult to draw reliable conclusions. Therefore, achieving uniform stiffness and thickness will ensure reproducible results across different batches and experimental setups. Key outputs for this criterion include the effective elastic modulus values for various PDMS substrate concentrations and precise measurements of their thickness.

## Biocompatibility

In the context of biomaterials used in biomedical and cell culture applications, biocompatibility ensures that the material supports normal cellular functions, does not induce immune responses or toxicity, and allows for desired interactions with biological components without compromising their integrity or functionality.

The sterilization protocol must be highly effective in eliminating all forms of microbial contamination, ensuring that the PDMS substrates remain sterile throughout the experimental period. To verify sterility, DMEM containing 10% FBS, 2% L-glute, 1% P/S will be added to the sterilized substrates within the wells and incubated at 37°C for 5 days. The absence of contamination will be confirmed by monitoring for any signs of microbial growth over the 5 day incubation period [64]. This will be evidenced by the lack of turbidity, absence of colour change, and no observable microbial colonies in the media [65]. These results will demonstrate that the sterilization procedure is effective, ensuring that the PDMS substrates provide a contamination-free environment suitable for subsequent cell culture experiments.

Coating the surface with biomolecules is expected to successfully render the PDMS substrates hydrophilic, as evidenced by water contact angle measurements. This modification is expected to promote robust cell attachment and growth, crucial for facilitating successful cell culture experiments.

Following sterilization and hydrophilicity enhancement, an AlamarBlue assay is conducted to assess the biocompatibility of the PDMS substrates. This assay aims to evaluate the proliferation and viability of cells cultured under various substrate stiffness conditions. It is hypothesized that cells cultivated on PDMS substrates will exhibit comparable viability and proliferation rates to those typically observed in cells cultured on standard plastic culture plates. This confirmation will serve to validate the cellular viability status of cells cultured on sterile PDMS substrates. Key outputs of this criterion encompass verifying sterility through pH colour change confirmation, assessing hydrophilicity via contact angle measurements, and evaluating cellular viability and metabolic activity using AlamarBlue analysis.

## Usability

High usability ensures that the protocol or system enhances productivity, reduces errors, and promotes a positive user experience during cell evaluation on PDMS substrates. The protocol is expected to successfully eliminate or significantly reduce substrate autofluorescence, thereby allowing clear and accurate fluorescent imaging of cells on the PDMS substrates. Autofluorescence can interfere with fluorescence-based assays and imaging, leading to background noise and reducing the clarity of cellular signals.

Moreover, the protocol is designed to be versatile, accommodating a wide range of cell types including primary cells and established cell lines. This versatility is essential for enhancing the utility of PDMS substrates across various research disciplines and experimental contexts. It is anticipated that the PDMS substrates will support the attachment, proliferation, and functionality of multiple cell types, thereby demonstrating the robustness and adaptability of the protocol. This adaptability not only facilitates standardized testing across different cell types but also enables comparative analyses within and

between studies, thereby supporting meta-analyses. Key outcomes of this criterion involve achieving successful cellular immunofluorescence on PDMS surfaces and demonstrating cellular versatility.

The protocol developed in this study addresses these concerns by providing a comprehensive protocol for PDMS fabrication for use in cell culture.

## Results

### Mechanical stability

Ensuring the consistency in stiffness and thickness of the PDMS substrates is crucial to avoid variations in cell behaviour, which can obscure the reliability of the experimental outcomes. To this end, various concentrations of Sylgard 184 and Sylgard 527 were prepared and analysed. Sylgard 184 samples were mixed in ratios ranging from 2:1–10:1 of Part A to Part B, while Sylgard 527 samples were mixed in ratios ranging from 1:0.65 to 1:1.25. The mixing and curing processes were repeated on three separate days, with Sylgard 184 substrates cured for 24 hours and Sylgard 527 substrates cured for 48 hours. Each sample was manually poured into the wells, ensuring that the same mass was placed in each well to maintain consistency.

**Stiffness.** After curing, the PDMS substrates were subjected to microindentation to determine their effective elastic modulus ($E_{eff}$). The raw microindentation data was analysed on the DataViewer Software V2.5.0 (Optics 11, the Netherlands) and is available as a supporting document S1 Dataset with this article. $E_{eff}$ was calculated from the loading curve using the Hertz contact model, applying a depth limit of 16% of the probe tip radius (i.e., 8 µm for a 50 µm probe tip radius). Indentations were excluded if the probe was in contact with the sample before the test, if the probe did not find the surface of the sample, or if the $E_{eff}$ R² value was < 0.9, as these were not considered successful indents. The microindentation results are presented in Figs 1 and 2.

Statistical analysis was performed using GraphPad Prism Software V10.3.0.507 (GraphPad Software, La Jolla California USA, GPS-2590652-T). Within excel, outliers were first identified and removed by calculating the interquartile range (IQR) for each dataset. The upper and lower limits were determined by adding 1.5 times the IQR to the third quartile and subtracting 1.5 times the IQR from the first quartile, respectively. Data points falling outside these limits were excluded from further analysis. Within GraphPad, the Shapiro-Wilk normality test was used to identify the most suitable testing requirements. Given that the data were not normally distributed, the non-parametric Kruskal-Wallis ANOVA test was used to compare $E_{eff}$ across samples prepared on different days. In this study, statistical significance was defined as a p value < 0.05. All data were presented in Figs 1 and 2 as median [25th– 75th percentiles] with violin plots.

Out of the 20 indents performed per concentration for Sylgard 184, all samples tested had 20 successful $E_{eff}$ indent values. A probe with stiffness 180 N/m and tip radius 50 µm was used for all Sylgard 184 samples. Analysis showed no significant differences between samples mixed on different days, indicating that the mixing technique described in this protocol effectively produces PDMS substrates with consistent stiffness (Fig 1).

Out of the 30 indents performed per concentration for Sylgard 527, the 1:0.75 S1 and 1:1.25 S1 samples had 29 successful $E_{eff}$ indent values, while all the other conditions had 30 successful $E_{eff}$ indent values. A probe with stiffness 0.5 N/m and tip radius 50 µm was used for all Sylgard 527 samples. Similar to Sylgard 184, the analysis revealed no significant differences between samples mixed on different days, further confirming the effectiveness of the mixing techniques in producing PDMS substrates with consistent stiffness (Fig 2). While concentrations below 1:0.65 were tested, these formulations failed to fully cure, preventing sterilization and cell seeding.

**Thickness.** The thickness of the PDMS substrates was determined through a precise process involving z-height measurements using the Chiaro Nanoindenter. Initially, a matrix scan was performed across a well of an empty plate and the z-height of the probe was measured. Subsequently, the z-height of the probe across the substrate samples was measured by performing a matrix scan. The difference between these measurements (probe z-height on

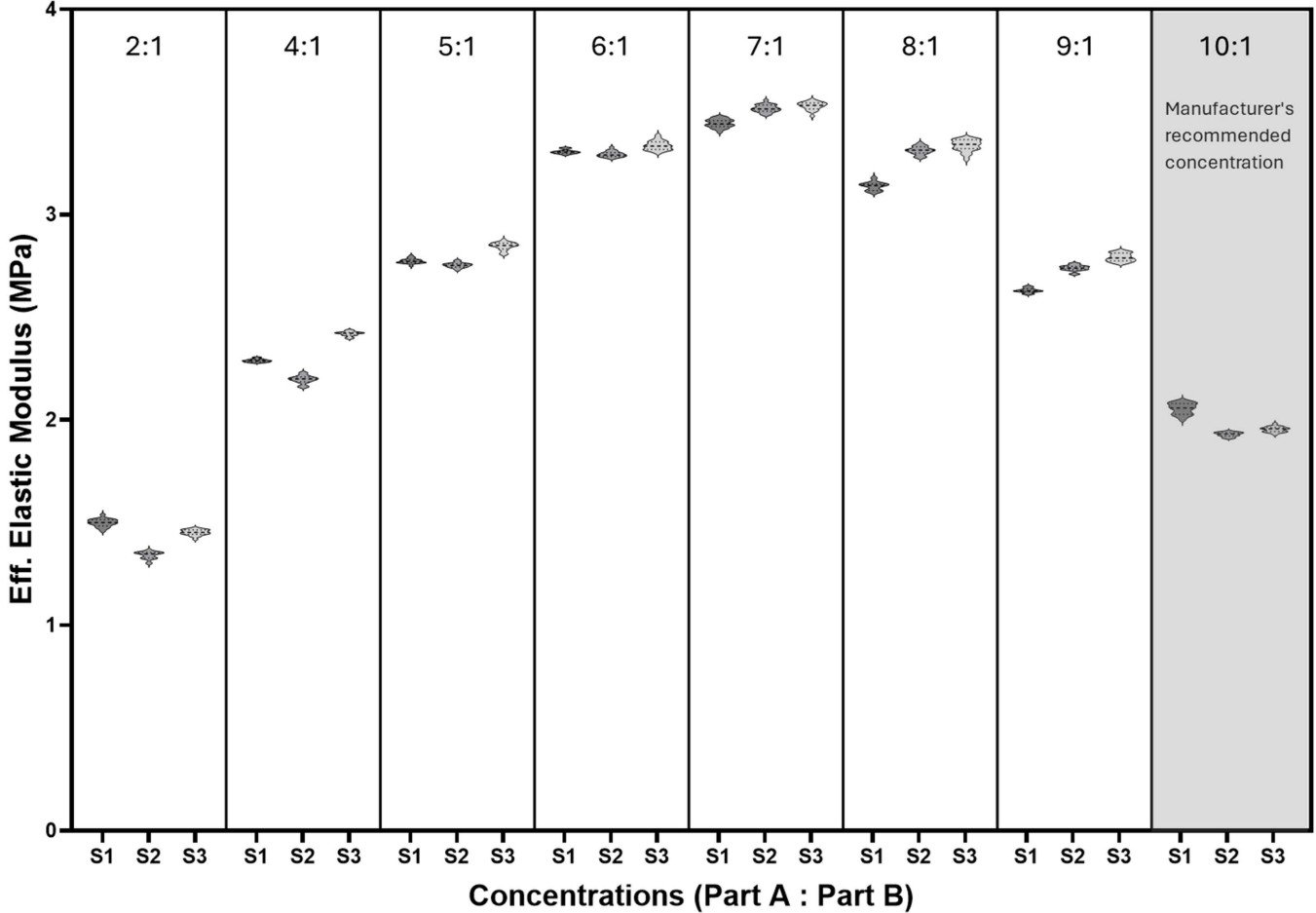

**Fig 1. E$_{eff}$ values for Sylgard 184.** E$_{eff}$ values for various Sylgard 184 concentrations (Part A: Part B) are presented, with three independent samples (S1, S2, and S3) prepared at different times. No statistically significant differences were observed among the samples within each concentration. The 10:1 ratio, shaded in grey, represents the manufacturer's recommended concentration.

substrate samples minus the average z-height on the empty plate well) provided the substrate thickness. Based on the manufacturer's recommendations, two PDMS formulations, Sylgard 184 at a 10:1 ratio and Sylgard 527 at a 1:1 ratio, were analysed. To ensure consistency, three wells for each concentration were cast at different times and assessed. This approach ensured accurate thickness determination.

The z-height data obtained from the nanoindenter revealed statistically significant differences between the samples cast at different times. This statistical significance arises primarily from the combination of a low standard deviation within each group and the relatively large sample size (n = 30 per group), which increases the power of the statistical test to detect even very small differences.

While the test detects a difference, the tightly clustered distributions within each sample group suggest that the actual variation in PDMS thickness is minimal. This indicates that the statistical significance is likely an artifact of high test sensitivity rather than a meaningful difference in thickness of the samples. Such effects are common when within-group variability is small, as even slight systematic variations can produce statistically significant results without practical relevance.

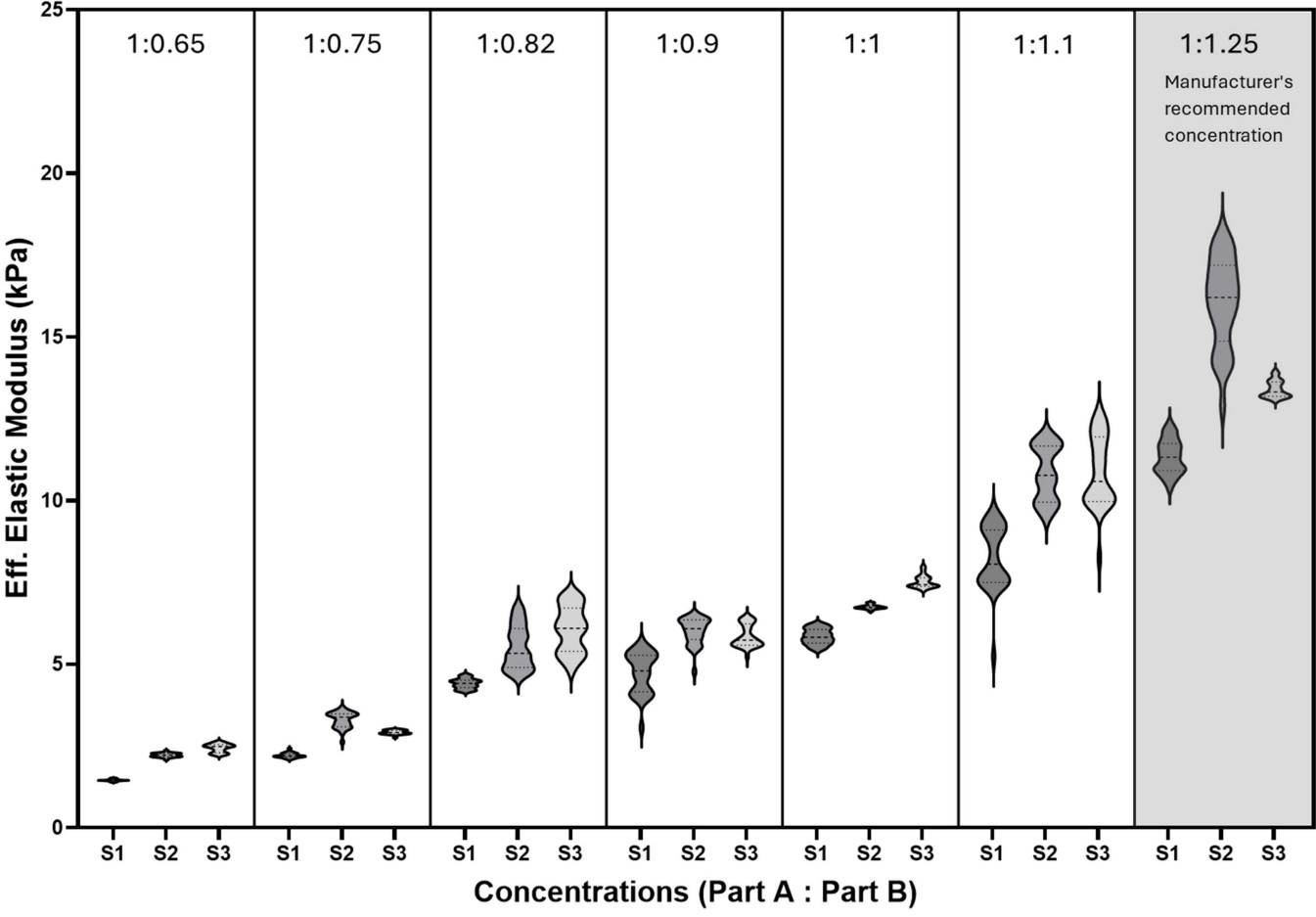

**Fig 2. E$eff$ values for Sylgard 527.** $E_{eff}$ values for various Sylgard 527 concentrations (Part A: Part B) are presented, with three independent samples (S1, S2, and S3) prepared at different times. No statistically significant differences were observed among the samples within each concentration. The 1:1 ratio, shaded in grey, represents the manufacturer's recommended concentration.

Importantly, the compact distribution of data points within each sample group confirms that thickness variations across individual samples are minimal, further supporting the flatness and uniformity of the films. This consistency underscores the reliability of the casting process and highlights the suitability of these substrates for cell culture applications, where surface uniformity is essential. Fig 3 presents the thickness measurements, and the raw data can be found in Supporting Document S2 Dataset accompanying this article.

Similar to the statistical analysis performed for the stiffness data presented previously, outliers were first removed in excel. Subsequently, the Shapiro-Wilk normality test was used to identify the most suitable testing requirements. Given that the data were normally distributed, the Brown-Forsythe and Welch ANOVA test was used to compare the gel thickness across samples casted at different times. In this study, statistical significance was defined as a p value < 0.05. The data are presented in Fig 3 as violin plots.

Overall, while the use of standard laboratory weighing scales and manual casting methods may introduce minor variations in gel thickness, these differences are minimal and within acceptable limits for most cell culture applications. The variability observed in thickness is acceptable when using free-pouring methods, as it does not significantly affect the stiffness of the gels when pouring between 1.34 and 2.23 grams of gel into a 6-well plate for cell culture (See Supporting

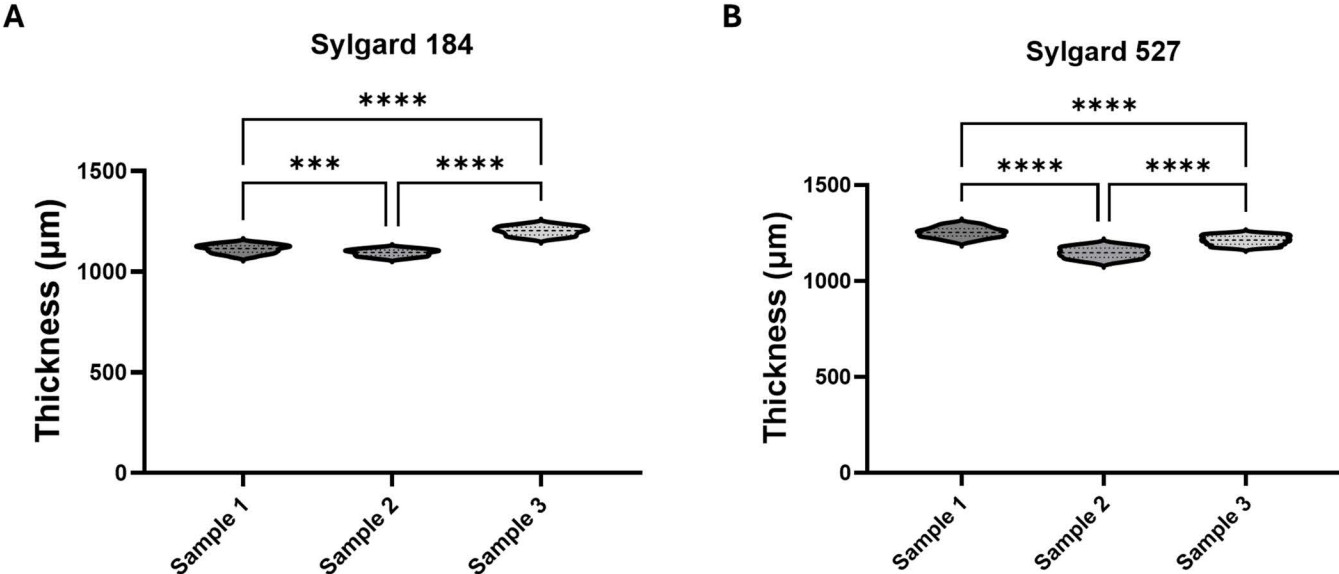

**Fig 3. Gel thickness for Sylgard 184 and Sylgard 527.** Analysis of gel thickness for (A) Sylgard 184 and (B) Sylgard 527, showing significant differences between samples cast at different times. Despite the statistical significance observed due to low within-group variability and a large sample size (n = 30 per group), the tightly clustered distributions indicate minimal actual variation in PDMS thickness. This suggests that the detected differences are likely a result of high test sensitivity rather than meaningful deviations in sample thickness.

Document S4 File and S3 Dataset). Although more precise methods, such as using highly accurate weighing scales or spin coating, could reduce these variations, such specialized equipment is often costly and not accessible in all laboratory settings. The protocol presented here is designed to be both accessible and reproducible across a wide range of laboratories, ensuring consistent results without the need for expensive or specialized equipment.

### Biocompatibility

**Hydrophilicity.** Polydopamine (PDA) has been recently exploited to improve cell behaviour on various substrates [33,34,58,59,63]. Polydopamine was chosen as the coating for PDMS substrates to enhance their hydrophilicity due to its established ability to form robust, adherent films on PDMS substrates. The improved wettability facilitates the adsorption of biomolecular coatings such as collagen, poly-L-lysine, or fibronectin, which are critical for promoting cell adhesion, proliferation, and function. The strong adhesive properties of PDA enable tailored biochemical functionalization, making it a versatile strategy for optimizing cell-material interactions. Contact angle measurements confirmed that PDA-coated and PDA-collagen-coated substrates exhibited significantly lower contact angles compared to uncoated PDMS, indicating superior hydrophilicity (Fig 4). These results unequivocally support the use of polydopamine as the optimal coating for achieving the desired hydrophilic modification of PDMS substrates. The raw data is available in Supporting Document S4 Dataset accompanying this article.

Statistical analysis was performed by first removing outliers in Excel. Subsequently, the Shapiro-Wilk normality test was used to identify the most suitable testing requirements. Given that the data were normally distributed, the Welch's t test was used to compare the contact angles of water droplets on uncoated PDMS versus 0.01% PDA-coated PDMS. In this study, statistical significance was defined as a p value < 0.05. The data are presented in Fig 4 as a bar graph depicting the mean values with standard deviation error bars.

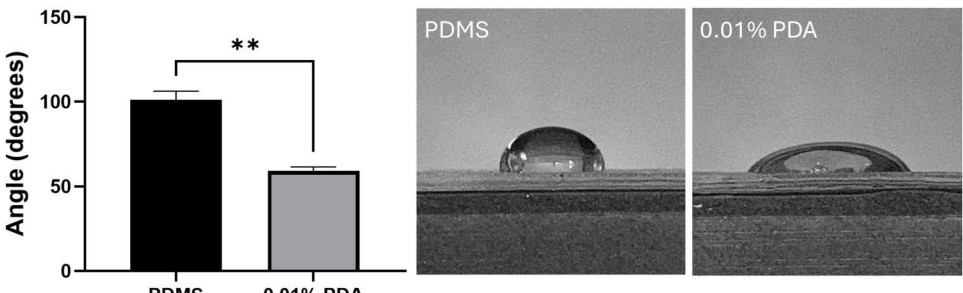

**Fig 4. Contact angle measurements for uncoated PDMS and 0.01% PDA-coated PDMS.** The results demonstrate a significant difference between the two conditions (p ≤ 0.0021), with the uncoated PDMS exhibiting a contact angle of ~100°, whereas the 0.01% PDA-coated PDMS exhibited a contact angle of ~60°. A 100 µL droplet of water was applied to both surfaces for measurement.

**Sterility.** The sterilization process utilized in this study involved spraying the PDMS substrates with 70% ethanol, followed by ethanol incubation under UV light, and subsequent washing with sterile deionised water before applying sterile filtered PDA to the surface of the gels. The sterility tests confirmed that the PDMS substrates coated with PDA remained uncontaminated throughout the incubation period. Media added to the sterilized substrates showed no signs of microbial growth, as evidenced by the absence of turbidity and colour change (Fig 5), and no colonies were observed upon further microscopic examination. These findings validate the efficacy of the sterilization protocol, ensuring that the substrates can be used in cell culture without the risk of contamination. The sterile environment provided by the PDMS substrates is critical for maintaining cell viability and ensuring the accuracy of experimental results, thereby supporting their use in various cell culture applications.

Statistical analysis was performed by first removing outliers in Excel. Subsequently, the Shapiro-Wilk normality test was used to identify the most suitable testing requirements. Given that the data were not normally distributed, the non-parametric Kruskal-Wallis ANOVA test was used to compare the decimal colour values of DMEM media between day 1 and day 5. In this study, statistical significance was defined as a p value < 0.05. The data are presented in Fig 5 as a violin plot accompanied by sample colour images. The raw data is available in Supporting Document S5 Dataset accompanying this article, while the supporting images are provided in Supporting Document S3 File.

**Cell viability.** Following sterilization and hydrophilicity enhancement, the biocompatibility of the PDMS substrates was assessed through performing an AlamarBlue assay and live/dead staining of cultured cells. The AlamarBlue assay demonstrated that cells cultured on PDMS substrates exhibited metabolic activity and viability rates comparable to those on standard plastic culture plates (Fig 6). High metabolic activity means that the cell is actively engaged in energy production, synthesis of molecules, and other life-sustaining activities. Conversely, low metabolic activity indicates reduced cellular function, which might be due to factors like disease, lack of nutrients, or exposure to toxins. The raw data for these findings are available in Supporting Document S6 Dataset.

Additionally, Live/Dead staining confirmed that multiple cell types were capable of growing on the PDMS gels (Sylgard 184 and Sylgard 527) without significant cell death, as indicated by the high percentage of live cells across all substrates (Fig 6). The corresponding raw data can be found in Supporting Document S7 Dataset. The minimal presence of dead cells, marked by propidium iodide staining, suggests that the PDMS substrates effectively support cell viability and proliferation, comparable to conventional plastic culture ware. These results highlight the biocompatibility of PDMS substrates, demonstrating their potential to maintain cellular functions and promote robust cell growth across various cell types. This finding is significant for applications in tissue engineering and biomaterials research, where PDMS is often used as a substrate material.

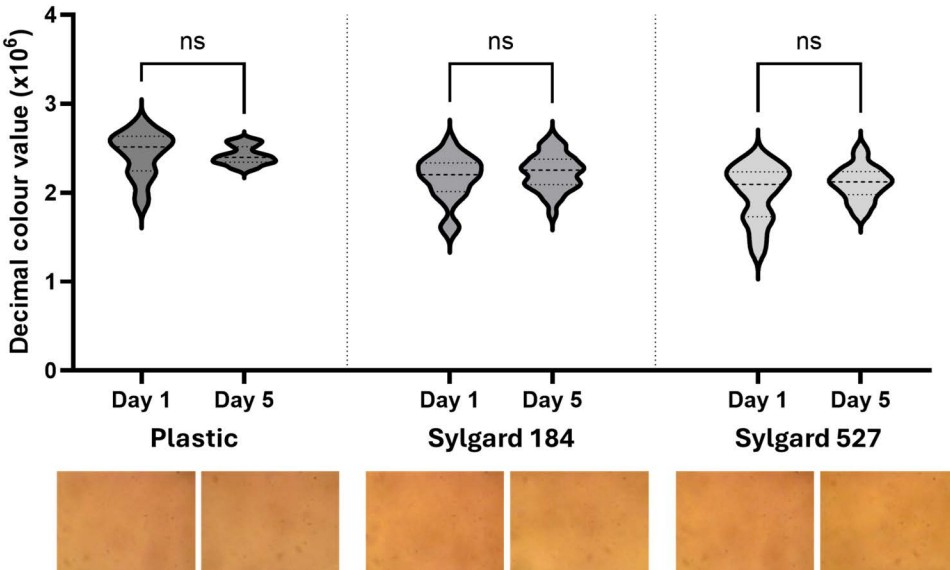

**Fig 5. Assessment of the sterility of PDA-coated PDMS substrates through colour change analysis over a 5-day incubation period.** RGB data was obtained using ImageJ. RGB values were converted to decimal format by calculating the product of the R, G, and B values (RxGxB). Data from days 1 and 5 was then compared for plastic, Sylgard 184, and Sylgard 527. No significant changes in media colour were observed over this time period, confirming sterility. Representative sample images for each condition are displayed below the corresponding data.

Statistical analysis was performed by first removing outliers in Excel. Subsequently, the Shapiro-Wilk normality test was used to identify the most suitable testing requirements. Depending on whether the data followed a normal distribution, either a Brown-Forsythe and Welch ANOVA or a non-parametric Kruskal-Wallis ANOVA was employed to compare the percentage reduction of Alamar Blue across plastic, Sylgard 184, and Sylgard 527. Statistical significance was defined as a p-value of < 0.05. The results are presented in Fig 6 as bar graphs displaying mean values with standard deviation error bars.

These results collectively affirm the effectiveness of hydrophilic coating, the suitability of the sterilization protocol, and the biocompatibility of PDMS substrates for cell culture applications. The combination of these characteristics underscores the potential of PDMS substrates as versatile biomaterials in biomedical and cell culture research.

### Usability

**PDMS autofluorescence.** The protocol developed for evaluating cells on PDMS substrates improved workflow efficiency and ensured dependable experimental results. It successfully minimized autofluorescence, leading to clear and high-quality fluorescent images with minimal interference (Fig 7). Autofluorescence, a common issue in fluorescence-based assays and imaging, can obscure cellular signals and introduce background noise, compromising data interpretation. This achievement highlights the protocol's efficacy in enhancing the clarity and reliability of fluorescent imaging, thus enabling precise visualization and analysis of cellular behaviour on PDMS substrates.

**Versatility.** The final objective was to demonstrate the application of the newly developed protocol to several different cell types. This capability is crucial for expanding the utility of PDMS substrates across various research disciplines and experimental contexts. As evidenced above by the Alamar Blue assay and Live/Dead staining results, multiple cell types remained viable on both Sylgard 184 and Sylgard 527 substrates. Furthermore, as illustrated in Fig 8, these PDMS substrates effectively supported the attachment, proliferation, and functionality of several types of cells: primary

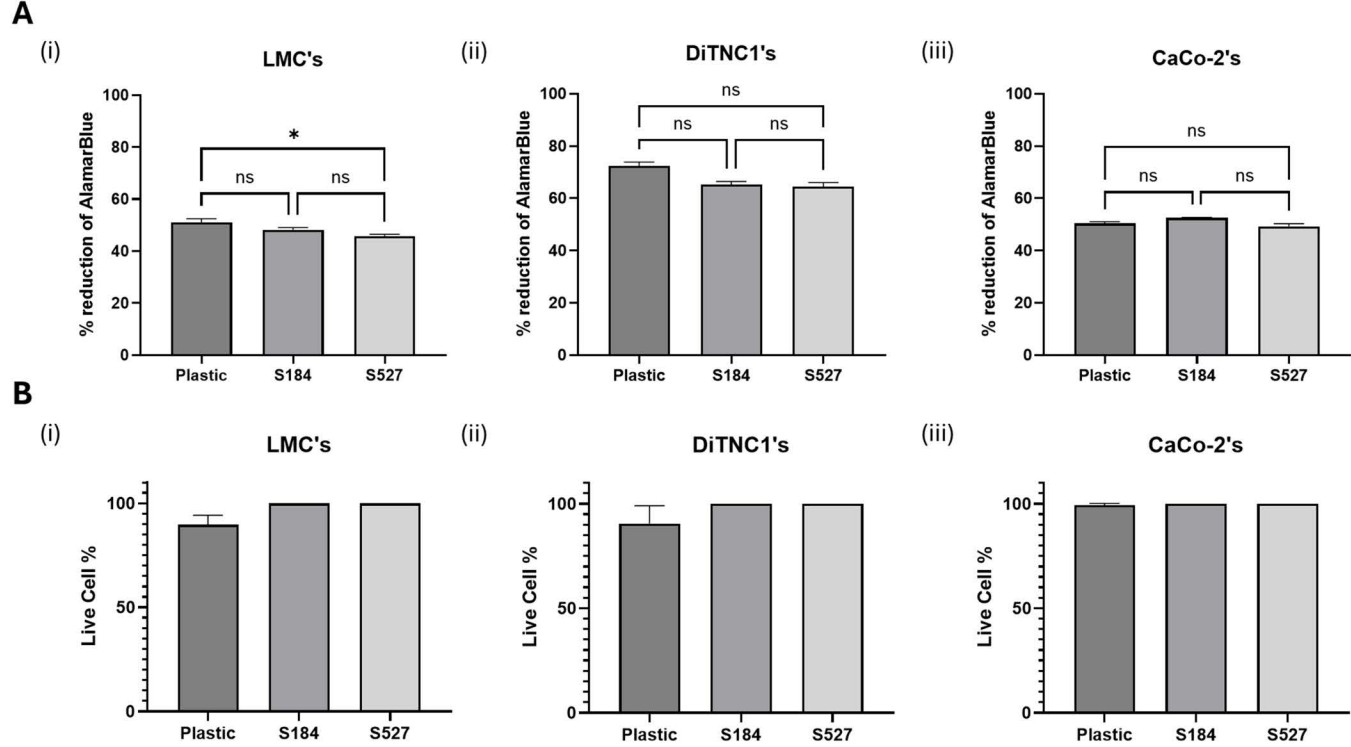

**Fig 6. Analysis of cell viability and metabolic activity and live/dead staining percentages.** (A) Percentage reduction of Alamar Blue for various cell types cultured on plastic, Sylgard 184, and Sylgard 527 substrates. All cell types seeded on Sylgard 184 and Sylgard 527 exhibited similar percentage reductions of Alamar Blue compared to plastic, indicating comparable cell viability and metabolic activity across these substrates. (B) Live/Dead staining results for cells cultured on plastic, Sylgard 184, and Sylgard 527 substrates. Calcein-AM was used to stain live cells (green fluorescence), while propidium iodide was used to stain dead cells (red fluorescence). Fluorescent images were captured after staining, with one well of each cell type pre-treated with 70% ethanol to induce cell death and validate the effectiveness of the dead cell stain. The number of live and dead cells was quantified using Cell-Profiler, and the percentage of live cells was calculated by dividing the number of live cells by the total number of live and dead cells.

human leptomeningeal cells, the rat astrocyte cell line DiTNC1, and the epithelial cancer cell line CaCo-2. The unedited images are provided in Supporting Document S2 Fig. The findings confirm that all these cell types adhered well and grew robustly on the PDMS substrates, showcasing their reliability and suitability for a wide range of cell culture applications without adversely affecting cell viability. This evidence highlights the potential of PDMS substrates to be a versatile and dependable choice for various cell-based experiments.

These results collectively highlight the efficacy and versatility of the protocol in enhancing experimental capabilities and supporting diverse cell culture applications on PDMS substrates. The protocol's usability, reduction of substrate autofluorescence, and compatibility with various cell types affirm its suitability for advancing biomedical and cell culture research.

## Discussion

The study presented a comprehensive protocol for fabricating PDMS substrates with tuneable stiffness for use in cell culture applications. The standardized method developed addresses several critical challenges that have historically limited the reproducibility and applicability of PDMS substrates in mechanobiology research. These challenges include achieving mechanical stability, ensuring biocompatibility, and enhancing the usability of PDMS substrates in diverse experimental contexts.

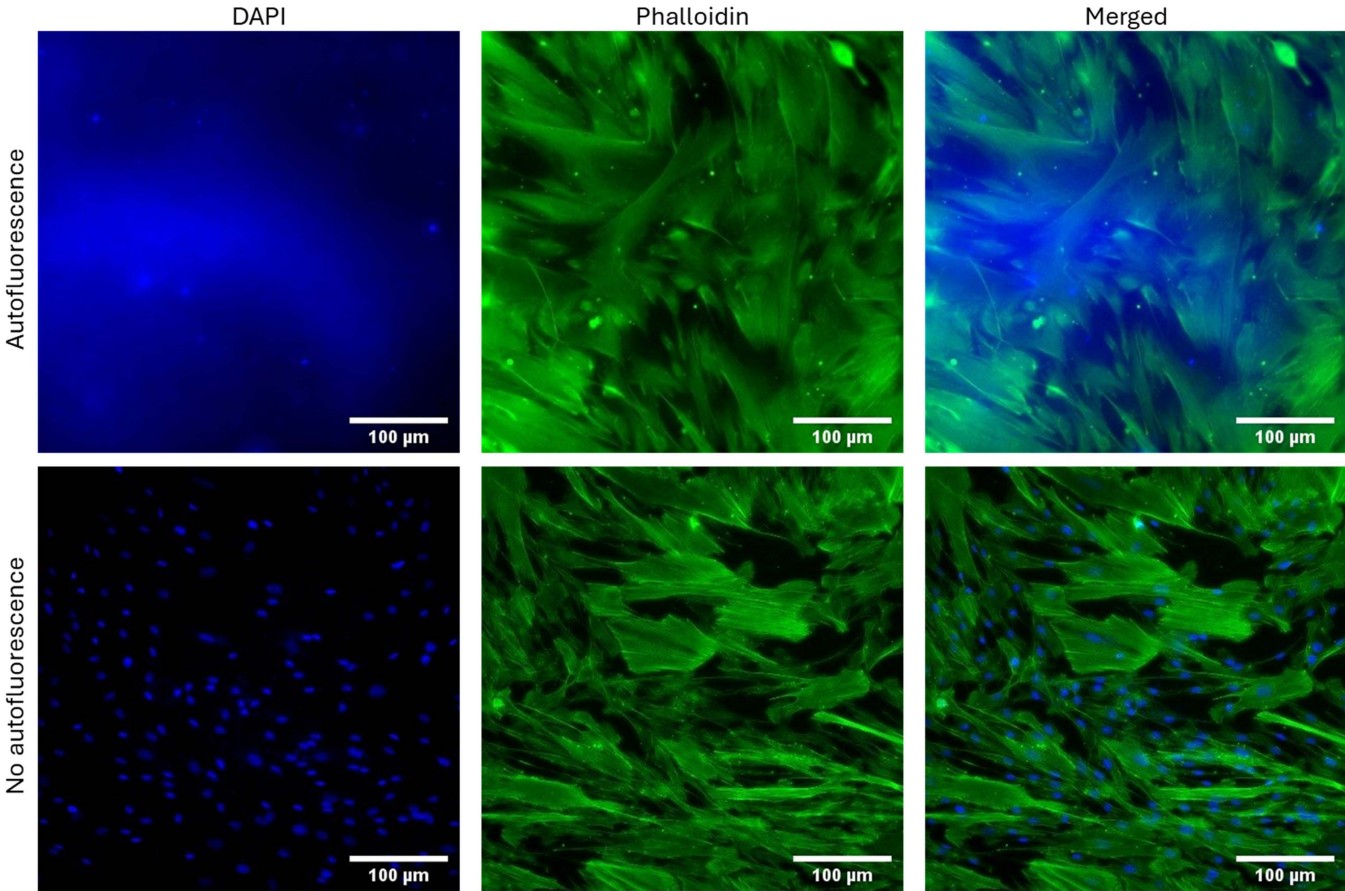

**Fig 7. Comparison of suboptimal and optimal fluorescent images of DAPI and phalloidin-stained leptomeningeal cells on Sylgard 184.** The suboptimal DAPI image resulted from insufficient staining time and inadequate washing, leading to poor signal quality. By increasing the DAPI incubation time and improving the washing protocol, higher quality images with reduced autofluorescence were obtained, demonstrating clearer nuclear staining. Scale bar = 100 µm.

## Mechanical stability

A significant strength of this protocol is its emphasis on mechanical stability, particularly the reproducibility of PDMS stiffness. By carefully controlling the mixing ratios and curing conditions, the study demonstrated that consistent mechanical properties could be achieved across different batches of PDMS substrates. To prevent thermal deformation of the plastic culture dishes, which could not withstand higher temperatures, PDMS samples were cured at 60°C. This curing process also reduced curing time and ensured uniform substrate properties. Maintaining this consistency is crucial, as even minor variations in substrate stiffness can significantly affect cell behaviour, including adhesion, proliferation, and differentiation [4,9,11–19].

For example, mixing the silicone base (Part A) with the curing agent (Part B) without a specialized mixer increases the likelihood of inconsistencies, underscoring the need for a robust, standardized manual mixing protocol. Following thorough mixing, degassing PDMS is crucial for removing trapped air bubbles and ensuring uniform material properties. The most common approach involves vacuum degassing, either using a vacuum chamber or an oven equipped with a vacuum pump [66–71]. However, alternative methods are available for laboratories without access to such equipment. Centrifugation can facilitate bubble removal by driving air pockets to the surface [72–74]. In the absence of both a vacuum chamber

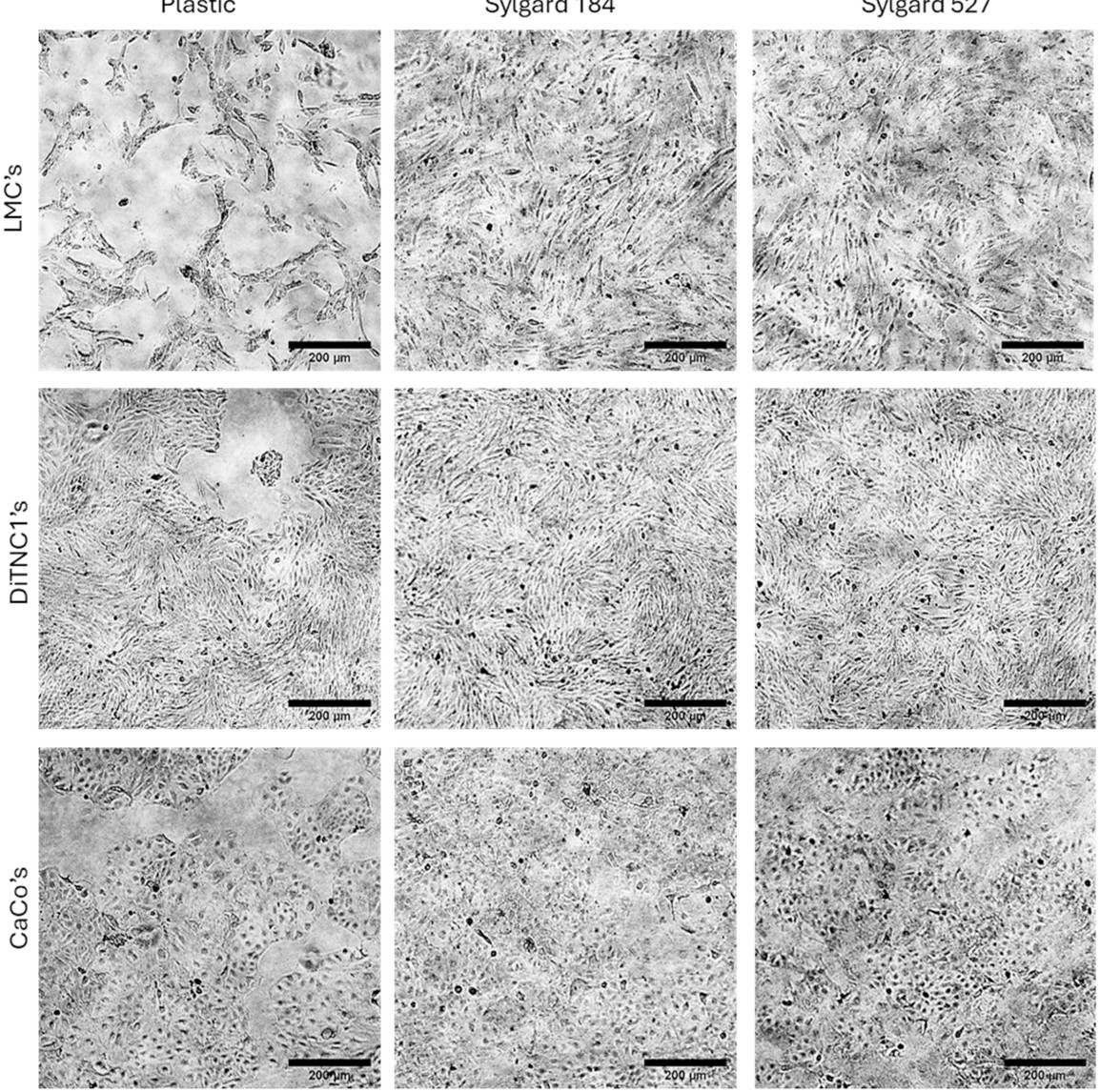

**Fig 8. Phase contrast images showing adherence and proliferation of various cell types on different substrates (Scale bar = 200 μm).** All three cell types exhibit strong adhesion and robust proliferation on Sylgard 184 and Sylgard 527, indicating the suitability of these substrates for cell culture applications.

and a centrifuge, allowing the mixture to rest after mixing can promote bubble dissipation. In addition to degassing, accurate weighing of the PDMS mixture before curing is essential, as the elastic modulus of PDMS membranes has been shown to decrease with increasing thickness. Additionally, the weighing of the mixture before curing is crucial; the elastic modulus of PDMS membranes has been shown to decrease with increasing thickness [75]. This implies that spin-coating PDMS onto glass coverslips for immunofluorescence analysis could result in cells being exposed to stiffer substrates than intended. Factors such as a higher base to curing agent ratio, longer curing time, and higher curing temperature can also lead to stiffer PDMS material [76,77].

The protocol's robustness, as evidenced by the lack of significant differences in stiffness and thickness across samples prepared on different days, underscores its reliability for mechanobiology research. This uniformity ensures that experimental outcomes are not confounded by variations in substrate properties, thereby enabling more accurate interpretations of how cells respond to mechanical cues.

## Biocompatibility

The biocompatibility of PDMS substrates is another critical aspect addressed in this study. The protocol's approach to sterilization and surface modification effectively mitigates the hydrophobic nature of PDMS, which typically impedes cell attachment and spreading.

Effective sterilization is crucial to prevent contamination in cell culture, as both residual chemicals and microorganisms can compromise experimental outcomes and cell viability. While traditional sterilization methods like autoclaving, dry heat, and gamma radiation are effective, they may alter the mechanical properties of PDMS. Therefore, this protocol emphasizes the use of alternative sterilization techniques, including ethanol immersion and UV irradiation, which do not affect the mechanical integrity of PDMS. Oxygen plasma sterilization is another viable method that maintains the mechanical properties of PDMS; however, the availability of this equipment can be limited in some laboratories [49–52].

To further enhance the biocompatibility of PDMS, surface modification strategies are necessary to mitigate its inherently hydrophobic nature. Various methods, including oxygen plasma treatment, UV/ozone treatment, chemical functionalization, and polydopamine (PDA) coating, can be employed to increase surface hydrophilicity. These methods, either alone or in combination with hydrophilic polymers like polyethylene glycol or polyvinyl alcohol (PVA), significantly enhance the biocompatibility of PDMS. Additionally, hydrogels or biomolecule coatings like gelatin, collagen, or fibronectin can be applied to further promote cellular interactions with the substrate [33,34,53–63].

In this study, PDA was selected as the surface coating to render the PDMS substrates hydrophilic due to its easy application. In cell culture applications, PDA coatings are typically prepared using dopamine concentrations between 0.1 mg/ml and 2 mg/ml. Lower concentrations, such as 0.1 mg/ml (0.01% w/v), have been shown to enhance cell adhesion, proliferation, and differentiation on PDMS substrates, with coating durations ranging from 1 hour to overnight, followed by rinsing and sterilization [78–81]. Similarly, certain cell types can proliferate effectively on higher PDA concentrations (>0.01% w/v), where thicker coatings enhance surface functionalization [82–85]. Thus, optimizing PDA concentration is essential for achieving the desired cellular responses. This study confirms that PDA enhances the hydrophilicity of PDMS substrates, as evidenced by reduced water contact angles, which is crucial for promoting robust cell adhesion. This improved wettability facilitates the efficient adsorption of biomolecular coatings, further supporting cell adhesion, proliferation, and function, key factors for successful cell culture experiments. The biocompatibility of PDA-collagen-coated PDMS seeded with cells was validated through AlamarBlue assays and live/dead staining, confirming that cell viability and proliferation were comparable to those on standard plastic culture plates. These results underscore the effectiveness of this approach in modifying PDMS surfaces to support cell viability and function.

## Usability

This study addresses the challenge of PDMS autofluorescence, a common issue that can interfere with fluorescence-based assays and imaging. To date, no studies have specifically addressed the challenges associated with staining PDMS substrates with fluorescent dyes. Casting the PDMS mixtures into plastic culture ware, rather than spin-coating them onto glass slides, often results in thicker PDMS substrates. This can cause fluorescent dyes to adhere more easily to the PDMS surface, complicating staining procedures. This protocol addressed these issues, enabling clearer and more reliable fluorescent imaging, which is crucial for accurate data interpretation in experiments involving immunofluorescence or other fluorescence-based analyses.

Moreover, the protocol's versatility was demonstrated by its compatibility with a wide range of cell types, including rat astrocytes (DiTNC1), epithelial cancer cells (CaCo-2), and primary human leptomeningeal cells (LMCs). While ISO 10993–5 specifies cell lines for cytotoxicity assessments, this study focused on establishing a protocol for modulating substrate stiffness to create physiologically relevant cell culture conditions. Therefore, primary human leptomeningeal cells, an immortalized cell line (DiTNC1), and a cancerous cell line (CaCo-2) were selected based on their relevance to the study objectives. This broad selection ensures that the method outlined in this study is applicable across different cellular contexts and provides insight into how various cell types adhere to and proliferate on PDMS substrates using this approach. The ability of PDMS substrates to support the attachment, proliferation, and functionality of diverse cell types broadens their applicability across various research disciplines. This adaptability not only facilitates standardized testing but also supports comparative studies, enhancing our understanding of cellular mechanotransduction.

This protocol offers a robust and accessible method for utilizing PDMS substrates in diverse research settings, ensuring reliable fluorescent imaging and compatibility with various cell types, while reducing the dependency on specialized equipment.

## Limitations and future directions

Despite the strengths of the protocol, several limitations and areas for future research should be considered. While the protocol effectively standardizes PDMS fabrication and addresses key challenges, one significant limitation is the lack of investigation into the long-term stability of the substrates under various environmental conditions, such as fluctuating humidity and temperature. These factors could potentially influence the mechanical properties, such as stiffness, and chemical stability of PDMS over extended culture periods. Future studies should explore the impact of these environmental variables on substrate performance, particularly in scenarios involving prolonged cell culture, to ensure the reliability and consistency of experimental results over time.

Furthermore, the protocol did not examine the effects of curing PDMS gels at different temperatures and durations. Since curing conditions are known to influence the cross-linking density and, consequently, the mechanical properties of the gels, such as stiffness, future work should systematically vary these parameters. By optimizing curing conditions, researchers could optimize the mechanical properties of PDMS substrates to meet specific experimental needs, potentially increasing the stiffness of Sylgard 184 beyond the maximum 4 MPa achieved in this study. This would allow for greater flexibility in adapting substrates to the requirements of different cell types or tissue models.

Another area for potential investigation is the analysis of a broader range of PDMS concentrations. The current study focused on a specific concentration range, but expanding this range in future experiments could provide a more comprehensive understanding of how varying the ratio of PDMS components affects substrate properties. For instance, exploring Sylgard 527 concentrations above 1:1.25 might result in higher stiffness values within the kPa range, better simulating the mechanical properties of stiffer tissues found in the body.

Additionally, while PDA coating was demonstrated to enhance the hydrophilicity of PDMS, thereby supporting the adsorption of biomolecular coatings, there remains an opportunity to explore alternative surface modification techniques. Investigating the combination of multiple surface treatments or the application of novel coating materials could further improve the performance of PDMS substrates. This is especially relevant for studies involving more challenging cell types or for the development of complex tissue models that demand highly specialized surface properties to support cell growth, differentiation, and function.

## Conclusion

This study provides a valuable and comprehensive protocol for the fabrication of PDMS substrates with tuneable stiffness, offering a reliable, reproducible, and versatile method that can be broadly applied in mechanobiology research. By addressing key challenges in PDMS substrate preparation, this protocol enhances the relevance of in vitro models to in vivo conditions, facilitating more accurate studies of cellular responses to mechanical cues. As the understanding

of cellular mechanotransduction continues to grow, protocols like the one developed in this study will be instrumental in advancing both basic research and the development of biomaterials for biomedical applications.

## Supporting information

**S1 File. Step-by-step protocol, also available on protocols.io.**
(PDF)

**S2 File. Step-by-step protocol.** Printable version for lab use.
(DOCX)

**S1 Fig. Process flow on PDMS fabrication protocol.**
(TIF)

**S1 Dataset. Raw and processed indentation data for Sylgard 184 and Sylgard 527 of various concentrations, including descriptive statistics.** S1, S2, and S3 correspond to indentation data from three separate wells, representing samples one, two, and three, respectively.
(XLSX)

**S2 Dataset. Raw and processed gel thickness data for Sylgard 184 (10:1) and Sylgard 527 (1:1), including descriptive statistics.** S1, S2, and S3 correspond to gel thickness data from three separate wells, representing samples one, two, and three, respectively. Gel thickness was determined by subtracting the Z-height of the gels, measured using the nanoindenter, from the average Z-height of the plastic well.
(XLSX)

**S3 File. Raw brightfield images of media colour used to assess sterility of the gels.** S1, S2, and S3 correspond to images from three separate wells, representing samples one, two, and three, respectively.
(DOCX)

**S4 File. Graphs of $E_{eff}$ values for various masses of Sylgard 184 (10:1) and Sylgard 527 (1:1).**
(DOCX)

**S3 Dataset. Raw and processed data of $E_{eff}$ values for various masses of Sylgard 184 (10:1) and Sylgard 527 (1:1).**
(XLSX)

**S4 Dataset. Contact angle data for uncoated PDMS and 0.01% PDA coated PDMS samples. Data was analysed using the contact angle plugin in ImageJ.**
(XLSX)

**S5 Dataset. Raw and processed data of decimal colour values for media incubated on sterilized Sylgard 184, Sylgard 527, and plastic over a 5-day period, including descriptive statistics. RGB data was obtained using ImageJ and converted to decimal format by calculating the product of the R, G, and B values (R×G×B).**
(XLSX)

**S6 Dataset. Raw and processed data on the percentage reduction of AlamarBlue, reflecting cell viability and metabolic activity.**
(XLSX)

**S7 Dataset. Raw and processed Live/Dead staining data for Sylgard 184 (10:1), Sylgard 527 (1:1), and plastic, including descriptive statistics.**
(XLSX)

**S2 Fig. Unedited images of the three cell types from Fig 8.**
(TIF)

**S3 Fig. Post-cured Sylgard 184.** Graph of Sylgard 184 cured at 60ºC (non-post cured) compared to post cured. After curing at 60°C for 24 hours, Sylgard 184 underwent a post-curing process at 200°C for 1 hour. The results indicate that stiffness increased with higher curing agent concentrations. In the corresponding data, the x-axis represents the curing agent concentration, expressed as the percentage of curing agent (Part B) relative to the base (Part A), while the y-axis denotes the effective elastic modulus.
(TIF)

## Author contributions

**Conceptualization:** Aisling J. Greaney, John J. E. Mulvihill.

**Data curation:** Aisling J. Greaney, Clíona M. McCarthy, Mannthalah Abubaker, Erin C. Reardon.

**Formal analysis:** Aisling J. Greaney, Clíona M. McCarthy, Frederick D. Crowley, John J. E. Mulvihill.

**Funding acquisition:** John J. E. Mulvihill.

**Investigation:** Aisling J. Greaney, John J. E. Mulvihill.

**Methodology:** Aisling J. Greaney, Clíona M. McCarthy, Jishnu Padacherri Vethil, Mannthalah Abubaker, Erin C. Reardon, Frederick D. Crowley, Eoghan M. Cunnane, John J. E. Mulvihill.

**Project administration:** John J. E. Mulvihill.

**Resources:** Jishnu Padacherri Vethil, Eoghan M. Cunnane, John J. E. Mulvihill.

**Supervision:** Eoghan M. Cunnane, John J. E. Mulvihill.

**Validation:** Aisling J. Greaney, Frederick D. Crowley.

**Visualization:** Aisling J. Greaney.

**Writing – original draft:** Aisling J. Greaney, John J. E. Mulvihill.

**Writing – review & editing:** Eoghan M. Cunnane, John J. E. Mulvihill.

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
