## [Decision Letter · Decision Letter 0]

26 Jan 2025

PONE-D-24-39286A Comprehensive Protocol for Polydimethylsiloxane (PDMS) Fabrication for use in Cell CulturePLOS ONE

Dear Dr. Mulvihill,

Thank you for submitting your manuscript to PLOS ONE. After careful consideration, we feel that it has merit but does not fully meet PLOS ONE’s publication criteria as it currently stands. Therefore, we invite you to submit a revised version of the manuscript that addresses the points raised during the review process.

We look forward to receiving your revised manuscript.

Kind regards,

Florian Rehfeldt

Academic Editor

PLOS ONE

“The research conducted in this publication was funded by the Irish Research Council under grant number GOIPG/2021/1433”

3. We note you have not yet provided a protocols.io PDF version of your protocol and/or a protocols.io DOI. When you submit your revision, please provide a PDF version of your protocol as generated by protocols.io (the file will have the protocols.io logo in the upper right corner of the first page) as a Supporting Information file. The filename should be S1_file.pdf, and you should enter “S1 File” into the Description field. Any additional protocols should be numbered S2, S3, and so on. Please also follow the instructions for Supporting Information captions [https://journals.plos.org/plosone/s/supporting-information#loc-captions]. The title in the caption should read: “Step-by-step protocol, also available on protocols.io.”

Please assign your protocol a protocols.io DOI, if you have not already done so, and include the following line in the Materials and Methods section of your manuscript: “The protocol described in this peer-reviewed article is published on protocols.io (https://dx.doi.org/10.17504/protocols.io.[...]) and is included for printing purposes as S1 File.” You should also supply the DOI in the Protocols.io DOI field of the submission form when you submit your revision.

If you have not yet uploaded your protocol to protocols.io, you are invited to use the platform’s protocol entry service [https://www.protocols.io/we-enter-protocols] for doing so, at no charge. Through this service, the team at protocols.io will enter your protocol for you and format it in a way that takes advantage of the platform’s features. When submitting your protocol to the protocol entry service please include the customer code PLOS2022 in the Note field and indicate that your protocol is associated with a PLOS ONE Lab Protocol Submission. You should also include the title and manuscript number of your PLOS ONE submission.

Reviewers' comments:

Reviewer's Responses to Questions

**Comments to the Author**

1. Does the manuscript report a protocol which is of utility to the research community and adds value to the published literature?

Reviewer #1: Yes

Reviewer #2: Yes

Reviewer #3: Yes

2. Has the protocol been described in sufficient detail?

To answer this question, please click the link to protocols.io in the Materials and Methods section of the manuscript (if a link has been provided) or consult the step-by-step protocol in the Supporting Information files.

The step-by-step protocol should contain sufficient detail for another researcher to be able to reproduce all experiments and analyses.

Reviewer #1: Partly

Reviewer #2: Partly

Reviewer #3: Yes

3. Does the protocol describe a validated method?

Reviewer #1: Yes

Reviewer #2: Yes

Reviewer #3: Yes

4. If the manuscript contains new data, have the authors made this data fully available?

Reviewer #1: Yes

Reviewer #2: Yes

Reviewer #3: Yes

**5. Is the article presented in an intelligible fashion and written in standard English?**

Reviewer #1: Yes

Reviewer #2: Yes

Reviewer #3: Yes

6. Review Comments to the Author

Reviewer #1: The authors report a lab protocol to standardize PDMS substrates with tunable stifness for mechaobiology studies for primary cells and cells lines. They discuss operational steps in terms of biocompatibility, mechanical stability,sterilization, autofluorescence and contact angle.

Some points to address:

-Introduce other ways to degas PDMS. In the protocol it is not clear PART A and PART B. Crosslinking agent is missing. Discuss differences between of the two Sylgard gels for applications.

-What is contact angle if you activate by oxygen plasma substrates before PDA?Why do you choose only PDA and concentration?

-How do you ensure flatness and uniformity of films in the same well?Explain better why there are statistical differences between samples.

-Which is the minimum uniform thickness you are able to obtain with casting on plates? Is it compatible with confocal analysis?

-Discuss the effect of curing temperature on elastic modulus.

Reviewer #2: The authors have described a protocol for fabricating PDMS with different stiffnesses for examining the effect of stiffness on cells. Some clarifications are required.

1. Was mechanical testing performed before or after sterilization and treatment with PDA and collagen? Do these processes have any effect on the mechanical properties of the surface on which the cells adhere?

2. Do the different PDMS formulations undergo relaxation during indentation? Is the material purely elastic or does it have a viscoelastic response?

3. In the protocol, it is mentioned that collagen is used to coat the surface but this is not included in the manuscript. Can the authors make it clear if collagen was used before adding cells. In the protocol, it also doesn't state what type of collagen is used or how it was dissolved (e.g. is it in acidic acid?)

4. More information or a reference is required for how you examined microbial growth. Did your media contain antibiotics or antimitotics? Were they incubated at 37C? How long was the test performed for?

5. For contact angle measurements after treatment with PDA, were these values consistent between all the different PDMS substrates? This data should be shown. Similarly was contact angle examined after collagen coating?

Reviewer #3: Authors developed a standardized method for fabricating PDMS substrates with tunable stiffness, ranging from kPa to MPa, suitable for diverse cell types using standard laboratory equipment in their manuscript. The protocol will be very useful for the cell culture audience. However, before publication some points need to be clarified.

- Authors should provide electron microscopy images before cell culture studies, surface morphology has an effect on cell behavior and should be proven that same for all samples.

- What is the stability of PDA coating please discuss

- Please provide data for PDS surface coverage percentages.

- Toxicity and proliferation data might give ideas about cell behavior. However, it does not represent that cells are healthy. Please use different words instead of cell health. More studies are needed to make this comment.

- Since this is a protocol for standardization, why did not the authors perform cell studies using a cell type from 10993-5 ISO

7. PLOS authors have the option to publish the peer review history of their article (what does this mean? ). If published, this will include your full peer review and any attached files.

**Do you want your identity to be public for this peer review?** For information about this choice, including consent withdrawal, please see our Privacy Policy .

Reviewer #1: No

Reviewer #2: No

Reviewer #3: No

---

## [Author Response · Author response to Decision Letter 1]

5 Mar 2025

Please see the uploaded document entitled "Response to Reviewers" for a more comprehensive response as well as the figures.

Reviewer #1:

The authors report a lab protocol to standardize PDMS substrates with tunable stifness for mechaobiology studies for primary cells and cells lines. They discuss operational steps in terms of biocompatibility, mechanical stability,sterilization, autofluorescence and contact angle.

Some points to address:

1. Introduce other ways to degas PDMS. In the protocol it is not clear PART A and PART B. Crosslinking agent is missing. Discuss differences between of the two Sylgard gels for applications.

o We agree that there are other applicable methods of degassing that can be used. I have made the following inclusions in the Discussion section that addresses the other methods of degassing that can be used in PDMS fabrication.

o Discussion, page 18, line 423:

“Following thorough mixing, degassing PDMS is crucial for removing trapped air bubbles and ensuring uniform material properties. The most common approach involves vacuum degassing, either using a vacuum chamber or an oven equipped with a vacuum pump [1–6]. However, alternative methods are available for laboratories without access to such equipment. Centrifugation can facilitate bubble removal by driving air pockets to the surface [7–9]. In the absence of both a vacuum chamber and a centrifuge, allowing the mixture to rest after mixing can promote bubble dissipation. In addition to degassing, accurate weighing of the PDMS mixture before curing is essential, as the elastic modulus of PDMS membranes has been shown to decrease with increasing thickness."

o I have changed the terms PART A and PART B in the protocols.io site in addition to what is mentioned in the manuscript.

o Protocols.io, Step 1:

“silicone base (Part A) and curing agent (Part B)”

o Protocols.io, Step 2:

“amounts of base (Part A) and curing agent (Part B)”

o Introduction, page 4, line 96:

“Many studies rely on specialized mixers for combining the silicone base (Part A) and the curing agent (Part B) of PDMS-based elastomers”

o Discussion, page 18, line 421:

“For example, mixing the silicone base (Part A) with the curing agent (Part B) without a specialized mixer”

o The Sylgards were chosen as they have similar chemistry with the main difference being the polymer chain lengths. The difference in chain length is the basis for the differences in mechanical strength (i.e. kPa vs MPa). Furthermore, Sylgard is a common and relatively cheap elastomer that any lab can get access to, which is one of the main objectives of this protocol.

o Introduction, page 3, line 58:

“Notably, PDMS-based elastomers, such as Sylgard 184 and Sylgard 527, are widely used in biological and engineering research due to their tuneable stiffness, ranging from the kilopascal (kPa) to the megapascal (MPa) scale, which allow them to be used as a mimetic of various biological tissue types. These differences arise from variations in polymer chain lengths and crosslinking densities, allowing researchers to modulate substrate mechanics while maintaining consistent surface chemistry [10]. This flexibility supports diverse applications; Sylgard 184, with a modulus in the MPa range, is suited for culturing cells that reside in rigid environments, such as fibroblasts, endothelial cells, and osteoblasts [11–14]. It is also widely used in microfluidics, soft lithography, and electronic encapsulation due to its durability and chemical resistance [15–17]. Sylgard 527, with a kPa-range modulus, is ideal for softer substrates used in stem cell and myoblast cultures [18–20] and is suitable for sealing and protecting various electronic devices, especially those with delicate components [21].”

2. What is contact angle if you activate by oxygen plasma substrates before PDA? Why do you choose only PDA and concentration?

o We do not have an oxygen plasma treater in our lab so looking at the contact angle after oxygen plasma treatment was not possible. Furthermore, we would stay using PDA as a technique as it aligns with the study objective of using widely available lab tools to carry out this protocol. However, we have attempted to address the comment by analysing other PDA concentrations that are commonly used for cell culture (i.e. 0.1 – 2 mg/ml).

o We tested different PDA concentrations on Sylgard 184 10:1 gels and found that there were no significant differences in contact angle between the three PDA concentrations and so choose to use 0.01% PDA for testing all cell types included in the paper in order to ensure the gels had good transparency for downstream immunofluorescent staining. All cell lines tested in this study adhered well using the 0.01% PDA concentration as well as 5ug/ml collagen coating seen from the data in Fig 10. Of course, the PDA concentrations would need to be optimised for one’s own cells to ensure cell adhesion. All PDA concentrations were left on the substrates at room temp for 24hrs.

o Results, page 13, line 298:

“Given that the data were normally distributed, the Welch’s t test was used to compare the contact angles of water droplets on uncoated PDMS versus 0.01% PDA-coated PDMS.”

o Discussion, page 20, line 462:

“In this study, PDA was selected as the surface coating to render the PDMS substrates hydrophilic due to its easy application. In cell culture applications, PDA coatings are typically prepared using dopamine concentrations between 0.1 mg/ml and 2 mg/ml. Lower concentrations, such as 0.1 mg/ml (0.01% w/v), have been shown to enhance cell adhesion, proliferation, and differentiation on PDMS substrates, with coating durations ranging from 1 hour to overnight, followed by rinsing and sterilization [22–25]. Similarly, certain cell types can proliferate effectively on higher PDA concentrations (>0.01% w/v), where thicker coatings enhance surface functionalization [26–29]. Thus, optimizing PDA concentration is essential for achieving the desired cellular responses. This study confirms that PDA enhances the hydrophilicity of PDMS substrates, as evidenced by reduced water contact angles, which is crucial for promoting robust cell adhesion.”

3. How do you ensure flatness and uniformity of films in the same well? Explain better why there are statistical differences between samples.

o The flatness and uniformity of the PDMS samples were assessed using topographical data obtained from nanoindentation. Since the topographical data from nanoindentation shows a consistent tilt in both the plastic dish and the PDMS samples, we can conclude that this tilt is due to the positioning of the well plate on the nanoindenter stage rather than an inherent unevenness in the samples. By accounting for the tilt observed in the plastic dish (comparing to the plastic control where no gels are applied), it becomes evident that the PDMS samples themselves are flat and uniform. We attempted to remove the effects of the tilt within the system but at that scale it is difficult to remove these slight tilt effects (i.e. 1-2˚can cause the images you see below).

Sylgard 184 10:1 Sylgard 527 1:1

Plastic

Sample 1

Sample 2

Sample 3

o Results, page 10, line 240:

“The z-height data obtained from the nanoindenter revealed statistically significant differences between the samples cast at different times. This statistical significance arises primarily from the combination of a low standard deviation within each group and the relatively large sample size (n = 30 per group), which increases the power of the statistical test to detect even very small differences.

While the test detects a difference, the tightly clustered distributions within each sample group suggest that the actual variation in PDMS thickness is minimal. This indicates that the statistical significance is likely an artifact of high test sensitivity rather than a meaningful difference in thickness of the samples. Such effects are common when within-group variability is small, as even slight systematic variations can produce statistically significant results without practical relevance.

Importantly, the compact distribution of data points within each sample group confirms that thickness variations across individual samples are minimal, further supporting the flatness and uniformity of the films. This consistency underscores the reliability of the casting process and highlights the suitability of these substrates for cell culture applications, where surface uniformity is essential. Figure 3 presents the thickness measurements, and the raw data can be found in Supporting Document S2 Dataset accompanying this article.”

o Results, page 11, line 275:

“Despite the statistical significance observed due to low within-group variability and a large sample size (n = 30 per group), the tightly clustered distributions indicate minimal actual variation in PDMS thickness. This suggests that the detected differences are likely a result of high test sensitivity rather than meaningful deviations in sample thickness.”

4. Which is the minimum uniform thickness you are able to obtain with casting on plates? Is it compatible with confocal analysis?

o The minimum uniform thickness achievable with casting on plates was determined to be 0.45 g (by mass as this is the most quantifiably consistent method of fabricating gels) of PDMS per well in a 6-well plate, as shown in Figure 4. This mass equates to a thickness of 0.463 mm in Sylgard 184 and 0.491 mm in Sylgard 527. Below this mass, it becomes challenging to achieve full coverage of the well bottom, making it difficult to form a uniform film thickness. The table below provides the theoretical thickness values corresponding to each mass of PDMS used.

Mass (g) Sylgard 184 thickness (mm) Sylgard 527 thickness (mm)

0.45 0.462 0.491

0.89 0.925 0.982

1.34 1.387 1.472

1.78 1.850 1.964

2.23 2.312 2.455

2.68 2.774 2.946

o Furthermore, the topography data for the minimum thickness, although showing a slight slant, aligns with the slant observed in the topography data from the plastic plate. This tilt is consistent and attributed to the positioning of the plate on the stage of the nanoindenter.

Sylgard 184 0.45g Plastic

o Regarding compatibility with confocal analysis, any of the obtained thicknesses would be suitable, provided the appropriate settings are used on the confocal microscope. Specifically, a z-stack approach can be employed to accurately locate cells on the PDMS substrate, ensuring that imaging can be performed effectively across different depths of the sample.

5. Discuss the effect of curing temperature on elastic modulus.

o Good point that is important to include when discussing PDMS fabrication. We include a sentence in the introduction about effects of curing temperature with references. Also, in the context of this study we have two criteria necessary when fabricating that dictate our choice of curing temperature; effect of temperature on plate (max 60 ℃) and consistency of curing. Please see the figure below to demonstrate the supplementary work we have carried to determine the effect of curing temperature, as well as post-cure temperature, on Sylgard 184 stiffness. We did not include this work as it was outside the remit of the study.

o Introduction, page 4, line 84:

“Sylgard 184 exhibits an increase in stiffness with higher curing temperatures due to enhanced crosslinking, with its elastic modulus ranging from ~1 to ~4 MPa between 25°C and 200°C [30,31]. In contrast, Sylgard 527 remains highly compliant, with negligible changes in elastic modulus, maintaining values within the low kPa range regardless of curing temperature [32,33].”

o Discussion, page 18, line 414:

“By carefully controlling the mixing ratios and curing conditions, the study demonstrated that consistent mechanical properties could be achieved across different batches of PDMS substrates. To prevent thermal deformation of the plastic culture dishes, which could not withstand higher temperatures, PDMS samples were cured at 60°C. This curing process also reduced curing time and ensured uniform substrate properties. Maintaining this consistency is crucial, as even minor variations in substrate stiffness can significantly affect cell behaviour, including adhesion, proliferation, and differentiation.”

o After curing at 60°C for 24 hours, Sylgard 184 underwent a post-curing process at 200°C for 1 hour. The results indicate that stiffness increased with higher curing agent concentrations. In the corresponding data, the x-axis represents the curing agent concentration, expressed as the percentage of curing agent (Part B) relative to the base (Part A), while the y-axis denotes the effective elastic modulus.

Reviewer #2:

The authors have described a protocol for fabricating PDMS with different stiffnesses for examining the effect of stiffness on cells. Some clarifications are required.

1. Was mechanical testing performed before or after sterilization and treatment with PDA and collagen? Do these processes have any effect on the mechanical properties of the surface on which the cells adhere?

o Great question and one we were curious about. We carried out the following experiment to address this question.

o Mechanical characterization was conducted prior to sterilization and subsequent treatment with PDA and collagen. The mechanical properties of PDMS samples were assessed under three conditions: uncoated (no PDA or collagen), PDA-coated, and PDA and collagen coated. No statistically significant differences were observed among these conditions for Sylgard 184 at a 7:1 concentration. However, for Sylgard 527 at a 1:0.82 concentration, statistical significance was detected (p < 0.05). Despite this, the observed stiffness variation (~5–7 kPa) falls within the range of batch-to-batch variability associated with differences in preparation time, as shown in Figure 2. Therefore, the statistical significance likely reflects methodological variability rather than a true difference in material properties.

2. Do the different PDMS formulations undergo relaxation during indentation? Is the material purely elastic or does it have a viscoelastic response?

o A valid question that we had already carried out but did not include in the original submission. As part of the indentation testing protocol, we do acquire elastic/viscoelastic response. We see from the figure below for Sylgard 184 that there is no relaxation over the course of the load being applied indicating that the material is not relaxing. However, there is some relaxation in the Sylgard 527 concentrations 1:0.65, 1:0.75, and 1:0.82 but this is expected for such soft substrates when the crosslinking agent is reduced. We argue that even for the lower concentrations of 527 that the drop in load is negligible in the kPa range.

Sylgard 184 Sylgard 527

2:1 1:0.65

4:1 1:0.75

5:1 1:0.82

6:1 ‘ 1:0.9

7:1 1:1

8:1 1:1.1

9:1 1:1.25

10:1

3. In the protocol, it is mentioned that collagen is used to coat the surface, but this is not included in the manuscript. Can the authors make it clear if collagen was used before adding cells. In the protocol, it also doesn't state what type of collagen is used or how it was dissolved (e.g. is it in acidic acid?)

o Good point that we left this key detail out, we initially left this out because our focus was primarily on rendering the gels hydrophilic, which was achieved using PDA. However, in hindsight, the collagen coating is an important aspect of the protocol. We include the below text in the results section to address this point.

o Results, page 12, line 281:

“The improved wettability facilitates the adsorption of biomolecular coatings such as collagen, poly-L-lysine, or fibronectin, which are critical for promoting cell adhesion, proliferation, and function. The strong adhesive properties of PDA enable tailored biochemical functionalization, making it a versatile strategy for optimizing cell-material interactions. Contact angle measurements confirmed that PDA-coated and PDA-collagen-coated substrates exhibited significantly lower contact angles compared to uncoated PDMS, indicating superior hydrophilicity (Fig. 4).”

o Discussion, page 20, line 471:

“This improved wettability

---

## [Decision Letter · Decision Letter 1]

6 Apr 2025

A Comprehensive Protocol for PDMS Fabrication for use in Cell Culture

PONE-D-24-39286R1

Dear Dr. Mulvihill,

We’re pleased to inform you that your manuscript has been judged scientifically suitable for publication and will be formally accepted for publication once it meets all outstanding technical requirements.

Kind regards,

Florian Rehfeldt

Academic Editor

PLOS ONE

Additional Editor Comments (optional):

Reviewers' comments:

Reviewer's Responses to Questions

**Comments to the Author**

1. Does the manuscript report a protocol which is of utility to the research community and adds value to the published literature?

Reviewer #2: Yes

Reviewer #3: Yes

2. Has the protocol been described in sufficient detail?

To answer this question, please click the link to protocols.io in the Materials and Methods section of the manuscript (if a link has been provided) or consult the step-by-step protocol in the Supporting Information files.

The step-by-step protocol should contain sufficient detail for another researcher to be able to reproduce all experiments and analyses.

Reviewer #2: Yes

Reviewer #3: Yes

3. Does the protocol describe a validated method?

Reviewer #2: Yes

Reviewer #3: Yes

4. If the manuscript contains new data, have the authors made this data fully available?

Reviewer #2: Yes

Reviewer #3: Yes

**5. Is the article presented in an intelligible fashion and written in standard English?**

Reviewer #2: Yes

Reviewer #3: Yes

6. Review Comments to the Author

Reviewer #2: The authors have addressed my concerns. I have no further comments on this manuscript.

Reviewer #3: Authors have made all necessary revisions in the submitted version. I suggest to publish the manuscript in Plos One.

7. PLOS authors have the option to publish the peer review history of their article (what does this mean? ). If published, this will include your full peer review and any attached files.

**Do you want your identity to be public for this peer review?** For information about this choice, including consent withdrawal, please see our Privacy Policy .

Reviewer #2: No

Reviewer #3: No

---

## [Editor Report · Acceptance letter]

PONE-D-24-39286R1

PLOS ONE

Dear Dr. Mulvihill,

I'm pleased to inform you that your manuscript has been deemed suitable for publication in PLOS ONE. Congratulations! Your manuscript is now being handed over to our production team.

Kind regards,

on behalf of

Dr. Florian Rehfeldt

Academic Editor

PLOS ONE